# Enteric Neuromyopathies: Highlights on Genetic Mechanisms Underlying Chronic Intestinal Pseudo-Obstruction

**DOI:** 10.3390/biom12121849

**Published:** 2022-12-10

**Authors:** Francesca Bianco, Giulia Lattanzio, Luca Lorenzini, Maurizio Mazzoni, Paolo Clavenzani, Laura Calzà, Luciana Giardino, Catia Sternini, Anna Costanzini, Elena Bonora, Roberto De Giorgio

**Affiliations:** 1Department of Veterinary Sciences, University of Bologna, 40064 Ozzano Emilia, Italy; 2Department of Medical and Surgical Sciences, University of Bologna, 40138 Bologna, Italy; 3IRET Foundation, 40064 Ozzano Emilia, Italy; 4Department of Pharmacy and Biotechnology, University of Bologna, 40126 Bologna, Italy; 5UCLA/DDRC, Division of Digestive Diseases, Departments Medicine and Neurobiology, David Geffen School of Medicine, UCLA, Los Angeles, CA 90001, USA; 6Department of Translational Medicine, University of Ferrara, 44121 Ferrara, Italy

**Keywords:** chronic intestinal pseudo-obstruction, enteric neuropathies, genes, neuro-myopathies, mitochondrial disorders

## Abstract

Severe gut motility disorders are characterized by the ineffective propulsion of intestinal contents. As a result, the patients develop disabling/distressful symptoms, such as nausea and vomiting along with altered bowel habits up to radiologically demonstrable intestinal sub-obstructive episodes. Chronic intestinal pseudo-obstruction (CIPO) is a typical clinical phenotype of severe gut dysmotility. This syndrome occurs due to changes altering the morpho-functional integrity of the intrinsic (enteric) innervation and extrinsic nerve supply (hence neuropathy), the interstitial cells of Cajal (ICC) (mesenchymopathy), and smooth muscle cells (myopathy). In the last years, several genes have been identified in different subsets of CIPO patients. The focus of this review is to cover the most recent update on enteric dysmotility related to CIPO, highlighting (a) forms with predominant underlying neuropathy, (b) forms with predominant myopathy, and (c) mitochondrial disorders with a clear gut dysfunction as part of their clinical phenotype. We will provide a thorough description of the genes that have been proven through recent evidence to cause neuro-(ICC)-myopathies leading to abnormal gut contractility patterns in CIPO. The discovery of susceptibility genes for this severe condition may pave the way for developing target therapies for enteric neuro-(ICC)-myopathies underlying CIPO and other forms of gut dysmotility.

## 1. Introduction

Severe gut dysmotility is a major impairment of intestinal propulsion due to alterations in various key cells, such as enteric neurons and/or glial cells, the interstitial cells of Cajal (ICC), and smooth muscle cells of the gastrointestinal (GI) tract [1,2,3]. In some patients, the functional impairment of digestive neuro-ICC-muscular-integrated systems is so pronounced that it hinders oral feeding, thereby leading to the patient’s poor quality of life and possible life-threatening complications. A typical example of this disabling condition is chronic intestinal pseudo-obstruction (CIPO), a clinical phenotype of severe gut dysmotility with a generally poor prognosis. Due to their underlying enteric neuro-ICC-muscular impairments, patients with CIPO manifest recurrent intestinal sub-occlusive episodes and clinical and radiological findings mimicking a mechanical gut obstruction [4,5]. CIPO patients complain of a variety of symptoms (e.g., chronic nausea, vomiting, abdominal distension, and constipation/diarrhea). After the acute sub-obstructive phases, during the inter-crisis period, enteral or parenteral nutrition is often required to prevent extensive weight loss and malnutrition [6]. In addition to adult cases, children can also be affected by intestinal pseudo-obstruction (hence, pediatric intestinal pseudo-obstruction—PIPO), presenting a clinical picture conferring high morbidity and mortality risks [7,8]. As in adult patients, the frequent manifestations and complications observed in PIPO involve chronic pain, small intestine bacterial overgrowth (related to gut dysfunction), and the occurrence of mid-gut or colonic (sigmoid) volvulus.

The lack of established animal models, which could provide a basis for a better understanding of the molecular pathways underlying CIPO, limits the development of effective treatments. Developing ad hoc, in vivo models will help to elucidate the altered mechanisms in the structure and/or function of the enteric nervous system (ENS), as recently shown for enteric glial changes in a subset of CIPO patients [9]. Highlighting the altered molecular mechanisms of CIPO is the prudent strategy by which to decipher its clinical complexity and guide pharmacological options or other targeted approaches for the development of personalized therapies and regenerative medicine (e.g., stem cell transplantation). The latter approach has been proven to be feasible through the recent progress in generating normal ENS development using human pluripotent stem cell (PSC)-derived neural crest cells (NCCs) and human intestinal organoids (HIOs) [10]. Notably, substantial advances have been made in disentangling the extensive network of enteric neurons and glial cells forming the enteric nervous system (ENS) [11,12] via the molecular definition of different classes of enteric neurons and non-neuronal cells. Therefore, identifying the molecular cause(s) of CIPO can elucidate the different defects in the ENS framework. Other basic studies will contribute to defining and correcting the molecular pathways through the smooth muscle layers as well as those of interstitial cells of Cajal (ICC).

The clinical phenotype of CIPO can be generated by mutations in different genes, indicating the high genetic heterogeneity underlying this disorder. As an example, CIPO is a feature associated with mitochondrial encephalomyopathies that can be caused by mutations in *TYMP*, a gene coding for thymidine phosphorylase, but also by polymerase gamma (*POLG*) mutations, or mutations in the mitochondrial DNA (mtDNA), which occur in mitochondrial encephalomyopathy with lactic acidosis and stroke-like episodes (MELAS) [13].

Herein, we will describe the genetic findings that have helped to unravel the novel molecular pathways involved in severe gut motility disorders such as CIPO and the impacts of technological advances in genomics and molecular phenotyping. In this conceptual framework, we will also address the recent identification of biallelic variants in the *LIG3* gene that result in a novel mitochondrial phenotype characterized by neuro-myopathic CIPO, brain leukoencephalopathy, and neuromuscular abnormalities. In a broader sense, the ultimate goal of this review is to provide the reader with a thorough discussion on the genetic and molecular alterations underlying severe gut dysmotility.

## 2. General Features of Gut Dysmotility in CIPO

As previously outlined, CIPO is a very severe form of gut dysmotility characterized by recurrent sub-obstructive episodes in the absence of any evidence of mechanical causes occluding the intestinal lumen [1,14]. The classification of CIPO is based on three major subtypes: (*i*) ‘‘secondary’’ forms, i.e., those cases related to a wide array of recognized pathological conditions; (*ii*) ‘‘idiopathic’’ forms, i.e., cases with unknown etiology; and, finally, (*iii*) “primary” forms, which can be applied to patients with a possible genetic origin. So far, the management of CIPO patients remains largely unsatisfactory, thus leading to frustration among the patients, their families, and physicians. In this review, we will specifically cover primary CIPO focusing on genetic abnormalities and the possible underlying neuro-muscular abnormalities. Combined clinical, histopathological, and genetic studies are crucial to identifying new perspectives in the understanding of CIPO, classifying the different forms of CIPO, and establishing correlations between histopathological and clinical phenotypes and causative genetic defects (summarized in Table 1). CIPO can be caused by abnormalities in the enteric neurons (neurogenic forms), smooth muscle abnormalities (myopathic forms) [15,16], and changes in the interstitial cells of ICC, of which the latter are very specialized non-neuronal/non-muscular cells endowed with a vast repertoire of function, the best-known being pace-making in the GI tract [17]. Notably, “neurogenic” forms of CIPO can also arise as a consequence of multisystemic disorders, such as diabetes mellitus and Parkinson’s, or in patients with Shy-Drager syndrome and other dysautonomic disorders [18].

In-depth in vitro and in vivo studies of gene variants are required to understand their impact in generating the severe enteric dysmotility experienced by CIPO patients. In this context, the discovery of the mutated genes represents the first step for developing novel targeted therapeutic strategies aimed at overcoming downstream molecular impairments.

## 3. CIPO with an Underlying Predominant Neuropathy

The neuropathological findings reported in neurodegenerative CIPO cases include various qualitative (neuronal swelling, intranuclear inclusions, axonal degeneration, and other lesions) and quantitative (oligoneuronal hypoganglionosis) abnormalities of the ENS [1,34]. Sporadic cases of enteric neuropathies are associated with two major patterns of abnormalities: (a) a marked reduction in intramural (especially myenteric) neuronal cells mainly associated with swollen neural cell bodies and processes, the fragmentation and loss of axons, and the proliferation of glial cells, and (b) a loss of normal staining in subsets of enteric neurons in the absence of dendritic swelling or glial proliferation [1]. Full-thickness specimens from patients with CIPO indicated that symptoms/clinical manifestations and severity increase as the number of enteric neurons decreases. Compared to control tissues, a 50% loss of neuronal cells in the myenteric and submucosal ganglia may be a “critical threshold” for recurrent sub-obstructive episodes, small bowel dilatation, and other severe symptoms [34]. Different findings have been identified in biopsies of the GI tract from patients with Parkinson’s disease (PD) or diabetes. Neuropathological and ICC changes have been detected in the gastric neuromuscular layer of patients with diabetic gastroparesis/gastroenteropathy [35]. Lewy pathology (i.e., intraneuronal deposits of phosphorylated α-synuclein) can be visualized in the ENS of patients with PD before a clinical diagnosis is established. This feature supports the notion that the GI tract exerts a key role in the pathogenesis of PD [36]. These abnormalities may likely involve the ENS and ICC more diffusely throughout the gut, but consistent data regarding CIPO related to diabetes mellitus and PD are still lacking. 

## 4. Genes Associated with Neuropathic Forms of CIPO: *RAD21* and *SGO1*

In recent years, our team and other groups have provided evidence of a genetic basis for the enteric neuronal degeneration and loss observed in specific forms of CIPO. The discovery of novel genes mutated in different patients represents the first step in identifying the cause of the downstream molecular impairment in CIPO. Homozygous mutations in *SGO1* and *RAD21*, which code for cohesin complex components, were identified in patients with CIPO [37]. Chetaille et al. described a new syndrome caused by a *SGO1* mutation affecting shugoshin-1’s structure and function. The authors defined this condition as chronic atrial and intestinal dysrhythmia (CAID) syndrome, i.e., a novel generalized dysrhythmia, indicative of the role of *SGO1* in mediating the integrity of human cardiac and gut rhythms, the latter being generated by ICC. Since shughoshin-1 (SGO1) is part of the cohesin complex, its dysfunction could result in consequences for long-range transcriptional regulation, possibly interfering with the expression of genes associated with CIPO [32]. Indeed, a recent study showed that an impaired inward rectifier potassium current, alterations of canonical TGF-beta signaling, and epigenetic dysregulation contributed to the development of the intestinal and cardiac manifestations observed in CAID syndrome [38]. 

We identified the homozygous causative variant in a large consanguineous family segregating an autosomal recessive form of CIPO in the Double-Strand-Break Repair Protein Rad21 Homolog (*RAD21*) gene. In the affected family members, other clinical features included a megaduodenum, a long-segment Barrett esophagus (up to 18 cm from the “Z-line”), and cardiac (interventricular septum) abnormalities of variable severity (OMIM 611376, also referred to as Mungan syndrome). We performed a whole-exome-sequencing analysis on the genomic DNA from two affected individuals and found a novel homozygous change, namely, c.1864 G>A in *RAD21,* responsible for damaging the missense variant p.Ala622Thr [30]. Any dysfunction of *RAD21*’s molecular structure and function can result in significant changes in many tissues, including the gut neuro-muscular layer. In fact, *RAD21* is part of the cohesin complex involved in the pairing and unpairing of sister chromatids during cell replication and division, as well as in regulating gene expression directly and independently of cell division [39]. The RAD21 subunit of the cohesin complex plays important structural and functional roles, as it serves as a physical link between the Structural Maintenance of Chromosome 1 (SMC1) and 3 (SMC3) heterodimers and the Stromal Antigen (STAG) subunit. RAD21 integrity regulates the association or disassociation of functional cohesin with chromatin and has a key role in double-strand break DNA repair [40,41]. In in vivo experiments using a zebrafish model, we recapitulated the CIPO phenotype observed in patients with the homozygous *RAD21* variant showing a severe impairment of motility and a marked reduction in the pan-neuronal marker HuC/D-immunoreactive enteric neurons, a finding reminiscent of an oligo-neuronal hypoganglionosis detected in CIPO patients [30]. Further studies using novel mouse models will help elucidate the molecular pathways altered by *RAD21* genetic defects.

A recent study by Le et al. (2021 [42]) showed that mutations in the genes coding for Erb-B2 Receptor Tyrosine Kinase 2 (*ERBB2*) and Erb-B3 Receptor Tyrosine Kinase 3 (*ERBB3*) led to a broad spectrum of developmental abnormalities, including intestinal dysmotility. A thorough gut histology assessment revealed aganglionosis, hypoganglionosis, and intestinal smooth muscle abnormalities in the affected patients. The cell type-specific *ErbB3* and *ErbB2* functions were determined through single-cell RNA sequencing data in a conditional *ErbB3*-deficient mouse model, which revealed a central role for *ErbB3* in enteric progenitors. Further mechanistic investigations will improve the understanding of the role of *ErbB3/ErbB2* pathways in ENS development, maintenance, and disease states [21,42]. 

## 5. CIPO with an Underlying Predominant Myopathy

Visceral myopathies are characterized by smooth muscle cell abnormalities and are most commonly caused by mutations in the contractile apparatus cytoskeletal proteins (for an extensive review, see [43]), such as *ACTG2* [16,20], *ACTA2* [19], *MYH11* [26,44], *MYLK* [28], *LMOD1* [25], *MYL9* [45], and *FLNA* [22,23,46]. Visceral myopathy (MIM# 155310) due to *ACTG2* pathogenic variants causes gut dysmotility due to smooth muscle dysfunction with phenotypes ranging from cases predominantly affecting the GI tract with typical CIPO features to the megacystis-microcolon-intestinal hypoperistalsis syndrome (MMIHS), which is characterized by severely disabling defects, such as dysfunctional bowels, reduced intrauterine colon growth (microcolon), and bladder-emptying defects requiring catheterization [43]. MMIHS can be considered a severe form of CIPO in early infancy with affected patients having a poor prognosis and short life expectancy. 

In patients whose bowels are mainly affected and a microcolon is absent, the condition can be labeled as myopathic CIPO. Causative heterozygous variants in *ACTG2* result in dominant disorders running in families or arising de novo in the affected subjects [20]. In a large study, the rate of molecular diagnosis in visceral myopathy cases was 64%, of which 97% was due to *ACTG2* variants. Missense changes in five conserved arginine residues of *ACTG2* contributed to 49% of cases [16]. The *ACTG2*-negative patients had a more favorable clinical outcome and more restricted disease. In the *ACTG2*-positive group, the poor outcome (i.e., total parenteral nutrition dependence, the need for transplantation, and death) was always associated with one of the arginine missense alleles. The analysis of the effect of the specific residues suggested the degree of severity of the missense changes, with substitutions at p.Arg178 exerting a more damaging effect than substitutions at p.Arg257 and p.Arg40, along with other less frequent variant alleles at p.Arg63 and p.Arg211. Four novel missense variants were also reported, including one transmitted according to a recessive mode of inheritance [47], indicating that the full genetic architecture of visceral myopathy has still to be fully characterized. Interestingly, in a recent review, Fournier and Fabre evaluated the mutation frequency observed in the genes involved in visceral myopathy in 117 published cases (112 patients and 5 pregnancies ended before birth) [48]. In concordance with previous studies, the most frequently reported mutations were in *ACTG2* (75/112, 67% of patients), *MYH11* (14%), and *FLNA* (13%).

It is worth noting that some patients with pathogenic *ACTG2* mutations develop the disease later in life and survive to adulthood without needing parenteral nutrition; conversely, others with the same *ACTG2* mutation may exhibit a severe form of the disease in childhood. Such variability in symptom severity in individuals with the same *ACTG2* gene defect strongly suggests that other factors, genetic or non-genetic, beyond the causative variants can impact the clinical phenotype. Understanding the reasons why clinical phenotypes vary despite identical *ACTG2* mutations may lead to new therapeutic strategies for these myopathic forms of CIPO.

The prevailing hypothesis is that *ACTG2* mutations cause smooth muscle abnormalities by disrupting the contractile cytoskeletal protein apparatus. In vitro studies in transfected cells showed an impairment of *ACTG2* polymerization and a reduction in smooth muscle cell contractility in the presence of the mutant form [28]. A recent study by Hashmi et al. demonstrated that in primary human intestinal smooth muscle cells (HISMCs) the *ACTG2^R257C^* mutation profoundly alters the *ACTG2* filament bundle structure, generating less robust fibers without altering the global actin cytoskeleton. Notably, *ACTG2^R257C^*-expressing HISMCs spread and migrated faster than the wild-type ones, suggesting that the mutation induces a less differentiated and less functional status in enteric smooth muscle cells [49]. 

Additional genes have been found to play a role in visceral myopathy pathogenesis. *ACTA2*, coding for a smooth muscle actin gene, is mutated in the multisystemic smooth muscle dysfunction syndrome (MSMDS; MIM #613834). The clinical features include bladder hypotonicity, abnormal intestinal peristalsis, and the prominent involvement of vascular and ciliary smooth muscles, leading to vascular aneurysms and mydriasis [19]. Different autosomal recessive forms of MMIHS are caused by biallelic loss-of-function variants in genes coding for proteins involved in actin–myosin interactions, such as *MYH11* (myosin heavy chain; [26], *MYLK* (myosin-light chain kinase; [28], *LMOD1* (leiomodin 1, an actin-binding protein expressed primarily in vascular and visceral smooth muscle [28]; and *MYL9* (regulatory myosin-light chain) [45]). Several studies have identified alterations in smooth muscle structural proteins and pathways related to smooth muscle function, providing substantial molecular insights into the disease’s pathogenesis. As an example, the loss of *LMOD1* in vitro and in vivo results in a reduction in filamentous actin, generating elongated cytoskeletal dense bodies and impairing intestinal smooth muscle contractility [28]. 

Mutations in the X-linked gene *FLNA* were previously only associated with forms of a neuropathic origin, but detailed immunohistochemical analysis has demonstrated diffuse, abnormal layering of the intestinal smooth muscle with no enteric neuron involvement [46]. *FLNA* has two isoforms—the longer one is predominant in the intestinal smooth muscle, and mutations in this isoform cause CIPO with periventricular nodular heterotopia in the brain [50].

Finally, myopathic forms of severe gut dysmotility have been identified in patients affected by myotonic dystrophy type 1, Duchenne muscular dystrophy, and oculo-gastrointestinal muscular dystrophy [51]. Scleroderma, systemic lupus erythematosus, dermatomyositis, polymyositis, amyloidosis, and ceroidosis can also cause myopathic CIPO [7].

## 6. Mitochondrial Disorders in Gut Dysfunction Related to CIPO

Mitochondrial neurogastrointestinal encephalopathy syndrome (MNGIE; OMIM # 603041) is an autosomal recessive disorder characterized by a severe impairment of GI motility (i.e., gastroparesis and CIPO) resulting from an underlying enteric neuro-myopathy [52] associated with peripheral neuropathy, ophthalmoplegia, and brain leukoencephalopathy that is detectable with magnetic resonance imaging [53]. MNGIE-related GI dysfunctions include bowel and gastric dysmotility leading to severe symptoms i.e., nausea, abdominal pain and distention, diarrhea, dysphagia, postprandial emesis, borborygmi, and gastro-esophageal reflux [54]. Patients usually present with progressive weight loss responsible for a thin body habitus and a severe reduction in muscular mass up to cachexia. The patients mainly die because of GI complications (e.g., perforation and hemorrhages) and an inadequate nutritional status [53,55]. Large, small bowel diverticula at the mesenteric border, likely secondary to the severe gut dysmotility present in the patients, were reported in many of the studied cases and are considered suggestive of the diagnosis [56]. Small bowel histological examination showed features characteristic of enteric myopathy such as a marked disfiguration of the outer layer (which exhibits significant degenerative vacuolization) and significant changes to the circular layer (also showing a marked reduction compared to the normal thickness), cytochrome-c-oxidase (COX) deficiency associated with mitochondrial proliferation, segmentary atrophy, and interstitial fibrosis. In contrast, the large bowel did not show major muscular changes. Other studies have shown that ICCs are deficient throughout the intestine of MNGIE patients [57]. Our group showed quantitative abnormalities (significantly reduced number) of enteric neurons in the small bowels of affected patients. Taken together, the morphological evidence indicates enteric neuro-ICC-myopathy as an underlying correlate of the severe dysmotility observed in MNGIE patients [55].

Several genes are involved in MNGIE etiology, e.g., *TYMP, POLG,* and *RRBM2. TYMP* codes for the thymidine phosphorylase (TP) enzyme, which regulates the mitochondrial nucleotide pool. TP is expressed in many human tissues, e.g., central and peripheral nervous systems, the GI tract, leukocytes, liver and platelets [58], and is a cytoplasmic enzyme that catalyzes the first step of mitochondrial dThd and dUrd catabolism by converting them to the nucleotides thymine and uridine, respectively, and 2-deoxy ribose 1-phosphate [57]. Thus, *TYMP* mutations determine a profound TP dysfunction leading to dThd and dUrd accumulation and a subsequent reduction in cytidine triphosphate (dCTP) in the plasma and tissues of MNGIE patients [33]. This imbalance affects mtDNA replication, causing mtDNA depletion and/or multiple deletions and point mutations as well as tissue damage (including the neuromuscular component of the GI tract) associated with the disease.

Based on the evidence that TP is highly expressed in the liver, our group pioneered the use of liver transplantation as a possible permanent source of TP replacement therapy in MNGIE patients [58]. So far, a number of patients with MNGIE have received liver transplantation worldwide, and this option, although requiring further supporting evidence, has been endorsed by the recently updated guidelines on MNGIE diagnosis and management [56]. 

Mutations in the gene coding for the catalytic subunit of mtDNA polymerase (*POLG*) are associated with a range of syndromes characterized by secondary mtDNA defects, including mtDNA depletion and multiple deletions (Mitochondrial DNA depletion syndrome 4B, MNGIE type; OMIM # 613662) [29]. The clinical manifestations closely resemble those of MNGIE, although brain leukoencephalopathy is not generally present. Finally, a recessive syndrome due to mutations in the gene coding for the ribonucleotide reductase-regulatory TP53-inducible subunit M2B (*RRM2B*) are associated with a clinical picture of ophthalmoplegia, ptosis, GI dysmotility, cachexia, and peripheral neuropathy, clinically overlapping MNGIE [31,59].

Apart from MNGIE, the histopathological analysis of other genetically driven mitochondrial disorders remains largely unresolved and studies in this area are necessary and eagerly awaited. Overall, the rarity of mitochondrial diseases in addition to their clinical variability, multisystemic involvement, and poor outcome significantly hinder tissue analysis and interpretation. Despite these limitations, genetic, functional, and morphological studies will likely unravel specific pathways for the development of new therapeutic approaches to the treatment of mitochondrial diseases.

## 7. A New Mitochondrial Recessive Disorder Associated with CIPO: Mutations in LIG3 

We recently characterized the gene defects underlying CIPO and neurological manifestations (reminiscent of MNGIE) in seven patients from three unrelated families [24]. In addition to CIPO, the most prominent and consistent clinical signs were neurological abnormalities, such as leukoencephalopathy, epilepsy, migraine, stroke-like episodes, and a neurogenic bladder. The DNA from these patients was subjected to whole-exome sequencing, which led to the identification of biallelic variants in the gene *LIG3*. All variants were predicted to have a damaging effect on the protein. The *LIG3* gene encodes the unique DNA ligase of the mitochondria and plays a crucial function in mtDNA repair and replication, in association with POLG. The study of the consequences of *LIG3* mutations in primary skin fibroblasts derived from patients and in transiently transfected cells expressing the different mutant vs. wild-type proteins revealed a severely reduced quantity of *LIG3* protein in the mutant cells, leading to a lack of ligase activity in the mitochondria compared to control fibroblasts with consequent defective mtDNA maintenance. The *LIG3* gene defects induced severe mtDNA depletion, but no accumulation of multiple deletions as observed in other mitochondrial disorders (e.g., MNGIE) altering the mitochondrial network [24] and leading to a severe imbalance in cell metabolism (i.e., impaired ATP production and increased mitochondrial reactive oxygen species generation). The resultant mitochondrial dysfunction underlies the clinical phenotype observed in these patients. In the gut, the histopathological analysis and neuronal HuC/D immunoreactivity demonstrated a significant loss of myenteric neurons in the colon, suggesting an underlying predominant neuropathy in the affected patients [60]. The ablation of *lig3* in the zebrafish model reproduced the brain defects and impaired gut transit together with an alteration in mitochondrial markers. Therefore, biallelic heterozygous loss-of-function variants in the *LIG3* gene result in a novel mitochondrial disease characterized by severe gut dysmotility, encephalopathy, and neuronal abnormalities.

## 8. Conclusions

In this review, we have described the identification of different genetic alterations that have given rise to the clinical picture of CIPO, which may reveal novel therapeutic strategies for patients with the enteric neuro-ICC abnormalities underlying this severe gut dysmotility. We showed evidence that a thorough genetic approach is a crucial step in highlighting the molecular pathways involved in ENS morpho-functional changes and thus enteric neuropathy, myopathy, and severe gut dysmotility. We indicate that a combined strategy based on accurate clinical phenotyping followed by histopathology and in-depth genetic analysis can reconstruct a model to better understand the neuro-(ICC)-muscular changes in CIPO. Next-generation sequencing is now allowing for the analysis of multiple genomic regions simultaneously, thereby shortening the time and costs of gene tests, and several studies have unveiled the presence of many independent genes for severe gut dysmotility. From these analyses and with the histopathological data available so far, three main phenotypes can be highlighted: (1) predominant neuropathy (e.g., *RAD21*-related) or ICC-related neuronal alterations (e.g., *SGO1*-related); (2) myopathy (e.g., *ACTG2*-dependent); (3) neuro-myopathy due to mitochondrial dysfunction (e.g., *TYMP*-, *POLG*-, and *LIG3*-related). Clearly, the definition of these three subsets requires further tests and research. Moreover, the discovery of additional genes will be crucial to identifying altered pathways with which to better understand the clinical variability of these complex syndromes/diseases. Changes to the neuro-myo-ICC functional aspects driven by genetic factors in severe dysmotility/CIPO are emerging as important and fascinating topics fuelling the interest for translational neurogastroenterology and the deciphering of novel therapeutic targets for patients with CIPO and related disorders.

## Figures and Tables

**Table 1 biomolecules-12-01849-t001:** Genes mutated in CIPO patients (in alphabetical order), with the relevant forms of the disease and associated key features.

Form of CIPO	Gene	Gene Name	Map Position	Key Features	References
*Myopathic*	*ACTA2*	Smooth muscle aortic alpha-actin 2	10q23.31	Multisystemic smooth muscle dysfunction syndrome	[19]
*Myopathic*	*ACTG2*	Enteric smooth muscle actin	2p13.1	Visceral myopathy 1, megacystis-microcolon-intestinal hypoperistalsis syndrome 5	[20]
*Neuropathic*	*ERBB2*	Erb-B2 Receptor Tyrosine Kinase 2	17q12	Visceral neuropathy, familial, 2, autosomal recessive	[21]
*Neuropathic*	*ERBB3*	Erb-B3 Receptor Tyrosine Kinase 3	12q13.2	Visceral neuropathy, familial, 1, autosomal recessive	[21]
*Myopathic*	*FLNA*	Filamin A	Xq28	Intestinal pseudo-obstruction, X-linked, congenital short-bowel syndrome	[22] [23]
*Mitochondrial*	*LIG3*	DNA Ligase III	17q12	Mitochondrial neurogastrointestinal encephalomyopathy	[24]
*Myopathic*	*LMOD1*	Leiomodin 1	1q32.1	Megacystis-microcolon-intestinal hypoperistalsis syndrome 3	[25]
*Myopathic*	*MYH11*	Myosin Heavy Chain 11	16p13.11	Megacystis-microcolon-intestinal hypoperistalsis syndrome 2 (AR), Visceral myopathy 2 (AD)	[26]
*Myopathic*	*MYL9*	Myosin Light Chain 9, regulatory	20q11.23	Megacystis-microcolon-intestinal hypoperistalsis syndrome 4	[27]
*Myopathic*	*MYLK*	Myosin Light Chain Kinase	3q21.1	Megacystis-microcolon-intestinal hypoperistalsis syndrome 1	[28]
*Mitochondrial*	*POLG*	DNA Polymerase Gamma, Catalytic Subunit	15q26.1	Mitochondrial DNA depletion syndrome 4B (MNGIE type)	[29]
*Neuropathic*	*RAD21*	RAD21 Cohesin Complex Component	8q24.11	Mungan syndrome (AR)	[30]
*Mitochondrial*	*RRM2B*	Ribonucleotide Reductase Regulatory TP53 Inducible Subunit M2B	8q22.3	Mitochondrial DNA depletion syndrome 8B (MNGIE type)	[31]
*Neuropathic*	*SGO1*	Shugoshin-1	3p24.3	Chronic atrial and intestinal dysrhythmia	[32]
*Mitochondrial*	*TYMP*	Thymidine Phosphorylase	22q13.33	Mitochondrial neurogastrointestinal encephalomyopathy (MNGIE); mitochondrial DNA depletion syndrome 1	[33]

**Notes:** AR = autosomal recessive; AD = autosomal dominant; CIPO = chronic intestinal pseudo-obstruction; p = short chromosome arm; q = long chromosome arm.

## Data Availability

Not applicable.

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
