# Peer review of "Enteric Neuromyopathies: Highlights on Genetic Mechanisms Underlying Chronic Intestinal Pseudo-Obstruction"

_biomolecules, 2022, doi:10.3390/biom12121849_

Round 1
Reviewer 1 Report
Chronic intestinal pseudo-obstruction (CIPO) is frequently associated with various disease syndromes, the etiology underlying CIPO is complex and remains largely unclear. The authors gave a comprehensive summary of the genes that have been found to be associated with the primary forms of CIPO. Overall, the authors have covered all the major findings related to CIPO. Nevertheless, the authors may need to reorganize the information as stated below to help the reader to follow the article. It is also important to better define and standardize the CIPO classification in the introduction section and throughout the article, respectively.
Specific comments:
1. In the abstract, what does “This translational approach” mean? Do the authors refer to "Discovery of susceptibility genes to CIPO" or "defining disease mechanism"? it is not translation approach.
2. Section 1: The introduction is confusing. “Herein, we will explain how genetic studies have helped to unravel novel molecular pathways involved in mitochondrial GI disorders”. It seems that the review article will only focus on mitochondrial GI disorders, but it is unlikely the case.
3. line 60, it is unclear why “the genetic heterogeneity underlying CIPO is given by mitochondrial encephalomyopathies”
4. The information in the introduction should be presented in order (similar to those in the abstract). It’s hard to follow what messages the authors want to deliver in the introduction.
5. The authors has classified the CIPO into “neuropathy”, “ICC”, “myopathy”. This information should be included in Table1. It is not clear how to define “mitochondrial CIPO”. Is it based on the genes or the disease involved? A clarification is needed. Also, should “mitochondrial-neuro-(ICC)-myopathy” be used throughout the paper?
6. Spell out the abbreviations when first time they were used: such as SMC1/SMC3, STAG.
7. Section 3: Have the authors published the data regarding the mouse mutant with Rad21 pAla526Thr? It seems to be inappropriate to include the unpublished data in a review article. If it has been published, please cite the corresponding article.
8. Section 4. Should it be part of “neuropathy”?
9. Sections 5 & 6 should be merged and reorganized.
10. Section 7, should the information presented in the second paragraph go first? an introduction of MINGIE is required before going into how it is related to CIPO.
11. Figure 1. In general, PPI is used to show the interactions among the genes. Nevertheless, the authors used it to conclude that there are three subtypes of CIPO. According to what the authors described about “mitochondrial-neuro-(ICC)-myopathy”, the PPI should show the mitochondrial genes are interacting with both neuropathy- and myopathy- associated genes. Thus, the figure is misleading.
Reviewer 2 Report
Dear authors,
I like the review and the writing. The focus on different genes and mutations is logical, and the given overview is sound.
I have only a couple of points to address or discuss:
1) You have only one reference from 2022, maybe there is some new work published till now?
2) Figure 1 is not really helping and the legend is very small. all coloured lines and gene clusters are a little confusing...
3) For the introduction, I would also be interested to read more about the involved cell types, as you claim to plan a therapeutic approach, the important cell types are very interesting. We work a lot with enteric neurons and glia, so there is maybe some potential.
4) You have 8 self-citation out of 46 references. Maybe you can also include some more literature from other groups, too?
Regards
Round 2
Reviewer 1 Report
The authors have fully addressed my concerns. The paper has been greatly improved and is suitable for publication.